# Impact of the COVID-19 and Socioeconomic Status on Access to Care for Otorhinolaryngology Patients

**DOI:** 10.3390/ijerph191911875

**Published:** 2022-09-20

**Authors:** Minju Kim, Jin-A Park, Hyunkyung Cha, Woo Hyun Lee, Seung-No Hong, Dae Woo Kim

**Affiliations:** 1Department of Otorhinolaryngology-Head and Neck Surgery, Seoul National University College of Medicine, Boramae Medical Center, Seoul 07061, Korea; 2Department of Otolaryngology, Kangwon National University Hospital, College of Medicine, Kangwon National University, Chuncheon 24289, Korea

**Keywords:** COVID-19, healthcare disparities, health services accessibility, socioeconomic status, otolaryngology, health insurance

## Abstract

Since December 2019, COVID-19 has greatly influenced public healthcare systems around the globe in various aspects, including limitation of healthcare accessibility due to lack of both human and financial resources, suspension of clinics, and fear of infection causing healthcare avoidance. The aim of this study was to investigate the impact of COVID-19 on access to healthcare for otorhinolaryngology patients from different socioeconomic status (SES) groups. Otorhinolaryngology patients’ disease severity status, diagnosed at the first hospital visit, was investigated during the pre -and post-COVID-19 pandemic era in a single medical center located in Seoul, Korea. An ordinal regression model was used to assess the impact of both SES and the COVID-19 pandemic on otorhinolaryngology diseases. Within the chronic rhinosinusitis group, lower SES was associated with a higher disease severity at the first visit compared to higher SES (OR = 3.25). During the COVID-19 pandemic, while the total number of outpatients was reduced, the severity of these ENT diseases seemed to increase compared to the pre-pandemic severity in every SES group. Our study demonstrates the negative impact a worldwide pandemic can have on healthcare inequity and disease severity, and highlights the importance of re-allocating fundamental resources for those in need during periods of public health crisis.

## 1. Introduction

The COVID-19 outbreak, first identified in December 2019 in China, was declared a global emergency on 30 January 2020, by the World Health Organization (WHO) [1] and as a pandemic on 11 March 2020. Social distancing and lockdown policies to control the public crisis may have prohibited people from seeking appropriate medical care. Furthermore, many individuals with pre-existing medical conditions were restricted from medical access [2]. According to self-reports, as of June 2020, 41% of United States adults postponed or avoided medical care due to concerns about COVID-19, even with 12% who presumably needed urgent medical care.

Equal access to healthcare services is an essential component of health rights, but it has been compromised during the pandemic and has resulted in healthcare inequities. Healthcare inequalities have always existed in a time of global crisis. In fact, reports of these inequalities exist worldwide and date back to the 1918 Spanish influenza. During this pandemic, one study from Norway showed mortality rates were highest among Oslo’s working-class residents [3]. During the H_1_N_1_ pandemic, high morbidity and high mortality due to this influenza-like illness were positively associated with lower income populations in the United States. Today, COVID-19 highlights the inseparable relationship between access to healthcare, social structure, and economic inequity [4]. In the annual State of Care report for England, published on 16 October 2020, The British Care Quality Commission (CQC) pronounced that COVID-19 was magnifying inequalities [5]. It has also been reported that, due to the pandemic and lockdown, the slum communities of Bangladesh, Kenya, Nigeria, and Pakistan have faced an inability to find healthcare for non-COVID-19 conditions [6]. Currently, it is assumed that the pandemic is disproportionately affecting socially disadvantaged individuals [7].

The concept of healthcare inequity is defined as discrepancies in health between advantaged and disadvantaged socioeconomic status groups [8,9]. Culer and Wagstaff have established four definitions of healthcare equity: equality of utilization, distribution according to need, equality of access, and equality of health [10]. Healthcare access is defined as the availability of health resources, various costs of using those resources, and the actual utilization of healthcare systems [11].

Otorhinolaryngology diseases are characterized by a broad spectrum of symptoms and physical features that may vary greatly in range and severity. Health disparities in the field of otorhinolaryngology are of particular interest, given that these problems can have a direct negative impact on quality of life due to functional impairment of the head and neck region. Hence, otorhinolaryngology diseases tend to be relatively more sensitive to factors that limit access to healthcare. In pediatric otorhinolaryngology, inequities in treating various diseases, including otitis media, acute rhinosinusitis, and sleep-disordered breathing have been reported [12]. Academic medical centers in the United States reported a drastic decrease in outpatient otorhinolaryngology cases during April 2020 as compared to inpatient surgical otorhinolaryngology cases [13]. In addition, emergency department consultation, including emergency cases such as vertigo, epistaxis, sudden hearing loss, otitis media, and peritonsillar abscess with otorhinolaryngology departments in Italian tertiary centers was reported to have decreased by 80.8% after the COVID-19 outbreak [14].

Prior to the pandemic, it had been reported that otorhinolaryngology patients with lower SES presented with higher disease severity on their first visit to hospital due to delayed healthcare-seeking behavior [15]. A similar association between disease severity at the time of initial evaluation and SES had also been reported with pulmonary hypertension in the United States [16].

The pandemic has demonstrated how the healthcare system can be challenged in guaranteeing healthcare equity. In the United States, Black and Hispanic children and those who were insured with Medicaid have been reported as less likely to attend medical appointments in otorhinolaryngology [17]. However, the impact of the COVID-19 pandemic on otorhinolaryngology healthcare disparities among groups with different socioeconomic status (SES) has not been thoroughly investigated.

For example, hearing loss is the most common disability around the globe according to the WHO [18]. The purpose of this study was to investigate the impact of COVID-19 on healthcare access for otorhinolaryngology diseases in different SES groups.

In this study, Otorhinolaryngology patients’ disease severity status at the time of their first hospital visit, and their medical health insurance status, were investigated during the pre- and post-COVID-19 pandemic era in a single medical center located in Seoul, Korea.

## 2. Materials and Methods

### 2.1. Study Design

A retrospective study was conducted from February 2018 to February 2021 using the medical records of patients from a metropolitan public hospital. Patients who visited the outpatient clinic and were diagnosed with chronic rhinosinusitis (CRS), sensorineural hearing loss (SNHL) and oral ulcers were enrolled. Subjects with craniofacial abnormalities, genetic anomalies, chronic medical diseases, and patients lost to follow-up before a proper diagnosis or severity evaluation was made were excluded. The timeline before and after the outbreak of COVID-19 was divided (Pre-COVID-19: 1 February 2018–29 February 2020; peri-COVID-19: 1 March 2020–28 February 2021). An ordinal regression model was used to analyze the impact of COVID-19 on access to healthcare in different SES groups. The Institutional Review Board of Seoul Metropolitan Government Seoul National University Boramae Medical Center approved this study (IRB #10-2020-15).

### 2.2. Socioeconomic Status

This study estimated a patient’s SES by adapting the health insurance coverage type governed by the Ministry of Health and Welfare. In 1977, mandatory social health insurance was introduced in Korea. It was introduced after the Law of National Health Insurance and Medical Care was passed and adopted. This law classifies the general population into two program groups according to income and socioeconomic status. The Medicaid program group represents the government-certified lowest-income group eligible for the National Basic Living Security Act, accounting for approximately 3% of all Koreans. National Health Insurance is mandatory for the group that includes the remaining general population covered by National Health Insurance. In this study, patients who were insured with Medicaid (Group 2) were compared to patients who were insured with National Health Insurance (Group 1) in order to analyze the impact of SES on healthcare access.

### 2.3. Severity of Diseases

In this study, healthcare accessibility was estimated by measuring the disease severity diagnosed by an otorhinolaryngology physician at the patient’s first outpatient clinic visit, following the assumption that lower SES can affect and reduce healthcare-seeking behavior, leading to lowered access to healthcare, and consequently causing a delayed appearance at the clinic.

Three otorhinolaryngology diseases which are prevalent and have relatively simple grading systems were included in this study: SNHL, oral ulcer, and CRS. Each disease group was divided into different severity groups according to standard guidelines. According to the World Health Organization Hearing Loss Grade, patients with SNHL were classified with mild (26–40 dB), moderate (41–60 dB), severe (61–80 dB), and profound or total (81 dB and above) loss according to the average pure tone thresholds at 500, 1000, 2000, and 4000 Hz. We classified oral ulcers according to the World Health Organization Oral Toxicity Scale as follows: grade 1: soreness ± erythema without ulceration; grade 2: erythema, ulcer, the patient can swallow solid food; grade 3: ulcers with extensive erythema, the patient cannot eat solid food; and grade 4: mucositis to the extent that alimentation is not possible [19]. CRS patients were divided into mild, moderate, and severe based on the Lund-Mackay Score of imaging workup as previously described (mild: score 0–8, moderate: 9–16, severe: 17–24) [20].

### 2.4. Statistical Analysis

All statistical analyses were performed using SPSS for Windows (version 20.0; IBM Corp, Armonk, NY, USA). Statistical significance was set at a *p* value less than 0.05 for double-sided comparisons for all statistical tests. The generalized estimating equation (GEE) method for matched ordinal multinomial was used to analyze the impact of COVID-19 on access to healthcare in different SES groups. The severity of diseases at the time of first hospital visit was examined, and the longitudinal change in disease severity was assessed for each group with ordinal regression analyses. Sex and age were matched between Group 1 (National Health Insurance) and Group 2 (Medicaid), and total double-numbered subjects were randomly selected from Group 1 for disease severity analysis.

## 3. Results

### 3.1. Demographics

A total of 2966 patients, visiting the outpatient clinic for the first time with common otorhinolaryngology diseases, were included in this study. Each disease group was composed of 989 (M:F = 483:506, age 52.78 ± 19.45) CRS patients, 1793 (M:F = 837:956, Age 69.63 ± 14.07) SNHL patients, and 184 (M:F = 75:109, Age 57.51 ± SD 18.45) oral ulcer patients. There was no sex difference between groups. However, the mean age of the SNHL group was higher than the other groups (*p* < 0.01).

### 3.2. Number of 1st Vist Patients during Peri-COVID-19 Period

In every disease group, the number of first-time patient visits decreased during the COVID-19 pandemic as compared to the number of first-time patient visits before the pandemic. In the CRS group, 31.9 patients first visited the hospital before the pandemic, and the number decreased to 17.1 patients per month after the pandemic. In the SNHL group, 51.9 patients per month first visited the hospital before the COVID-19 pandemic, then the number declined to 42.1 patients per month during the peri-COVID 19 period. The number of patients who presented with oral ulcers declined from 5.46 to 4.05 patients per month after the outbreak (Table 1).

### 3.3. Ratio of Disease Severity Pre and Peri-COVID-19

The overall disease severity of patients generally increased during the pandemic. (Table 2) The proportion of each disease severity group among disease groups was compared before and during the outbreak of COVID-19. In the CRS group, the proportion of moderate to severe disease increased during the peri-COVID-19 period in both Group 1 and Group 2 patients. Within the oral ulcer groups, the proportion of moderate to severe disease trended upwards after the pandemic, as in the CRS group. However, in the SNHL group, the proportion of profound hearing loss patients decreased during the peri-COVID-19 period in both SES groups.

### 3.4. Effect of Socioeconomic Status and COVID-19 on Disease Severity

For each disease, Medicaid (Group 2) was compared with age- and sex-matched controls (Group 1) who were randomly selected from the National Health Insurance patient group. (Table 2) Within the CRS group, the Medicaid subgroup showed higher disease severity (OR = 3.25, 95% CI, 1.859–5.676) compared to the National Health Insurance group, and disease severity increased during the peri-pandemic period compared to the pre-pandemic period (OR = 2.17, 95% CI, 1.142–4.135). In the SNHL group, the Medicaid subgroup showed higher disease severity (OR = 1.78, 95% CI, 1.259–2.539) compared to the National Health Insurance group, but unlike with other diseases, although not statistically significant, the disease severity tended to decrease during the peri-COVID-19 period. Patients with oral ulcers tended to have lower disease severity when they were insured with Medicaid, and this tended to increase during the peri-COVID-19 pandemic, although not statistically significantly (Table 3).

## 4. Discussion

Since 2019, the COVID-19 pandemic has challenged our public health system in many aspects. Healthcare resources, including human resources, intensive care units, and even personal protective equipment were redistributed in response to demand during the pandemic [21,22]. Therefore, screening and diagnostic procedures for common diseases have been limited or even suspended world-wide [23]. Furthermore, fear of COVID-19 infection, clinic closures, and social distancing policies have inhibited the public from seeking appropriate care for their pre-existing conditions [24,25]. Consequently, the avoidance or delay of appropriate medical care has likely caused an increase in morbidity in patients with preventable medical conditions [24]. This study demonstrates that the new coronavirus pandemic has increased risk for vulnerable patients with otorhinolaryngology diseases who already had limited access to healthcare.

In this study, healthcare accessibility was estimated by measuring the otorhinolaryngology disease severity following the assumption that individuals in the Medicaid program, the lower SES group, can have more difficulty seeking healthcare services, consequently causing delayed appearance at the clinic. It has been reported that markers of disease severity, such as bone erosion in allergic fungal rhinosinusitis, a subtype of chronic rhinosinusitis, are associated with rural areas, low economic status, and less access to healthcare [26]. In our previous study, which was conducted before the pandemic, we showed that people who were insured with Medicaid presented with higher severity on their first visit compared to those who were insured with National Health Insurance in CRS, SNHL and oral ulcer [15].

In our current study, as in the previous study, CRS (OR = 3.25) and SNHL (OR 1.78) patients who were insured with Medicaid tended to present with higher disease severity at the time of hospital visit, compared to the National Health Insurance group, before and during the pandemic. Notably for CRS patients, the impact of SES over disease severity decreased after the COVID-19 pandemic, which might be the result of general avoidance of healthcare visits due to the fear of the pandemic, especially when people were expecting to be in contact with other patients with upper respiratory symptoms. Also, before the pandemic, people who were insured with Medicaid might have sought healthcare less frequently than those with National Health Insurance did, and presented with higher disease severity on their first hospital visit. However, during the pandemic, even people who were insured with National Health Insurance used healthcare systems much less than before, resulting in a higher disease severity on their first visit. In fact, during the pandemic, factors such as fear of disease transmission and quarantine may have prevented people from seeking medical care in the early stages of their disease [23,24]. Insecure income during quarantine may have exacerbated the negative effects of SES on healthcare access during the outbreak [25], since people would have limited time for hospital visits especially for benign diseases as discussed in this study. Therefore, generally lowering the barrier to health care services across all socioeconomic groups should also be considered in public health polices during the era of pandemics.

For SNHL patients, the effect of SES on disease severity at the time of their first visit increased after the pandemic. As shown in the previous study, sensorineural hearing loss with lower SES was associated with higher disease severity before and after the pandemic [15]. However, disease severity tended to decrease during the pandemic, as the number of patients with profound hearing loss decreased. Hearing impairment has been proven to increase pre-existing economic inequality [27]. In this case, due to the pandemic, we assumed that people with disabilities, such as profound hearing loss, had restricted access to common healthcare resources [28,29]. Therefore, people with severe hearing loss may not have been able to visit medical centers and were excluded from the statistics on disease severity. Furthermore, to visit a secondary or tertiary medical center in Korea, an individual has to be referred from a local practitioner, which could be another blockage to seeking advanced healthcare in the time of pandemic.

The limitations of this retrospective study include a relatively short timeline and a small number of subjects. Therefore, a new study using an increased number of subjects that are recruited for a longer period of time might yield more comprehensible results. Additionally, patients with upper respiratory infection symptoms such as cough, rhinorrhea, and sputum (common symptoms in otorhinolaryngology patients), may have erroneously been transferred to the COVID-19 screening process first, resulting in falsely decreased visits to the outpatient clinic. Another limitation of this study is that disease severity can be affected by many other uncontrollable confounders [30]. However, many of these factors are also intimately associated with SES, which means their absence might yield similar results to this study. Also, health insurance status might not reflect various aspects of an individual’s socioeconomic status. Nevertheless, further discussion and investigations are needed in order to analyze the underlying causal factors and mechanisms of disparity involved in access to otorhinolaryngology care. A systematic method to assess and respond to exacerbated healthcare inequity, as well as informed distribution of healthcare assets during the time of pandemic, are necessary to our country’s healthcare system.

## 5. Conclusions

By examining the healthcare-seeking behavior of patients with otorhinolaryngology diseases, this study has demonstrated that there was a significant impact on healthcare inequity due to the pandemic. The COVID-19 outbreak seems to have affected the previously existing healthcare access discrepancies between different socioeconomic status groups with various conditions. Within the CRS and SNHL groups, the Medicaid group tended to have higher disease severity compared to the National Health Insurance group, and interestingly the OR decreased after the COVID-19 pandemic in the CRS group. However, in the SNHL group, the discrepancy between the two groups’ severity increased after the COVID-19 pandemic although not statistically significantly. In addition, patients with CRS and oral ulcers presented with higher disease severity at the time of the first hospital visit during the COVID-19 pandemic compared to the pre-pandemic era. It is essential to maintain efforts to provide fundamental medical services during the pandemic, especially with at-risk populations, in order to improve overall public health.

## Figures and Tables

**Table 1 ijerph-19-11875-t001:** Number of 1st visit patients per month during peri-pandemic era.

		NIH	Medicaid	Total
CRS	Pre-COVID	30.4/month	1.4/month	31.9/month
	Post-COVID	16.2/month	0.92/month	17.1/month
SNHL	Pre-COVID	47.5/month	4.3/month	51.9/month
	Post-COVID	38.4/month	3.68/month	42.1/month
Oral ulcer	Pre-COVID	4.79/month	0.67/month	5.46/month
	Post-COVID	3.48/month	0.57/month	4.05/month

CRS = Chronic rhinosinusitis, SNHL = Sensorineural hearing loss, OR = Odds Ratio. (Pre-COVID 19: 01 February 2018–29 February 2020, peri-COVID-19: 1 March 2020–28 February 2021).

**Table 2 ijerph-19-11875-t002:** Change in the proportion of disease severity during pre- and peri-COVID-19 periods.

		NHI	Medicaid	Total
CRS	Pre-COVID	Mild 70.1% Moderate 25.4% Severe 4.9%	Mild 40.3% Moderate 48.4% Severe 11.3%	Mild 60.7% Moderate 32.7% Severe 6.63%
	Post-COVID	Mild 45.5% Moderate 50% Severe 4.6%	Mild 18.8% Moderate 75% Severe 6.25%	Mild 27.1% Moderate 68.75% Severe 4.16%
SNHL	Pre-COVID	Mild 29% Moderate-Severe 46.8% Severe 15.6% Profound 8.7%	Mild 20% Moderate-Severe 41.8% Severe 21.8% Profound 16.36%	Mild 26.17% Moderate-Severe 45% Severe 17.7% Profound 11.2%
	Post-COVID	Mild 35.3% Moderate-Severe 42.4% Severe 16.5% Profound 5.9%	Mild 27.1% Moderate-Severe 37.5% Severe 22.9% Profound 12.5%	Mild 32.3% Moderate-Severe 40.6% Severe 18.8% Profound 8.3%
Oral ulcer	Pre-COVID	Mild 50% Moderate 47% Severe 2.9%	Mild 70.6% Moderate 17.6% Severe 11.8%	Mild 56.9% Moderate 37.3% Severe 5.9%
	Post-COVID	Mild 40% Moderate 60% Severe 0.0%	Mild 40% Moderate 40% Severe 20%	Mild 40% Moderate 53.3% Severe 6.67%

CRS = Chronic rhinosinusitis, SNHL = Sensorineural hearing loss. NHI = National Health Insurance.

**Table 3 ijerph-19-11875-t003:** The impact of socioeconomic status on disease severity before and during COVID-19.

		OR	95% CI of OR
CRS	SES	3.248	1.859–5.676
	COVID	2.173	1.142–4.135
SNHL	SES	1.787	1.259–2.539
	COVID	0.787	0.533–1.162
Oral ulcer	SES	0.675	0.226–2.013
	COVID	1.829	0.643–5.202

CRS = Chronic rhinosinusitis, SNHL = Sensorineural hearing loss, OR = Odds Ratio. Generalized estimating equation (GEE) method for matched ordinal multinomial was performed.

## Data Availability

The data that support the findings of this study are available from the corresponding author upon reasonable request.

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
