# Peer review of "Impact of the COVID-19 and Socioeconomic Status on Access to Care for Otorhinolaryngology Patients"

_ijerph, 2022, doi:10.3390/ijerph191911875_

Round 1

Reviewer 1 Report

Dear authors,

Thank you for the opportunity of reviewing your paper. The research topic is interesting. You use medical records data and correctly develop a set of descriptive analysis and regression analysis. I have no major comments regarding your methodology.

However, I believe that the sample size may prevent your results from being as generalizable and you claim. Additionally, it is not very clear the added-value of the paper relative to existing literature. Finally, I don’t think that the results that you are displaying are aligned with your research question.

I propose below a set of major and minor topics that, if properly addressed, might allow surpassing these issues and improve results robustness.

In my opinion, the major issues that should be addressed are:

1. Geography and External Validity: there is no indication in the abstract and in the introduction on the country / setting in which this analysis is being performed. This should be clear since the beginning of the paper – eventually, even in the title. Moreover, it should be discussed how are these results generalizable (or not) to other countries and other types of health systems: a discussion on the external validity of these results in missing in the paper.

2. Contribution: you claim that the concerns regarding patient access to healthcare during the pandemic have been already reported in the literature, with many different perspectives. Given this, and based on the paper, it is not clear what is the contribution of this paper to the current literature. I suggest you to clarify this in advance in the introduction (after referring to the existing literature) and in the discussion section. Clarifying your contribution, will help you “selling” your paper! In particular, it would be great to discuss why is it relevant to look at otorhinolaryngology patients. Do we expect results for this particular group of patients to be different? Why? What makes them special?

3. Research Question: I feel that the way you present your results is disconnected from your original research question. You claim that “The aim of this study was to investigate the impact of COVID-19 on access to healthcare for otorhinolaryngology patients from different socioeconomic status (SES) groups”. However, you present your results in terms of changes in the severity mix. You do not explicitly compare the access to care for the two groups – only the change in severity for the two groups. Thus, I advise you to rewrite your results, discussion and conclusion and align them more closely with your original (and relevant) research question. My next comments provide some suggestions to do this.

4. Results: you are mostly looking to severity mix changes and not to access to care. It would be extremely relevant to complement this with overall number of admissions. You provide the overall number of admissions for the total sample, but you do not distinguish this between groups. I suggest to i) provide additional information on the number of visits by groups of SES; ii) consider replacing figure 1 by a table and test whether the differences between groups are statistically significant. Include the information of the number of visits by groups in this table; iii) provide an explanation for the high volume of Medicaid patients presented in table 1 (earlier you claim that, on average, Medicaid patients represent 3% of the population. However, in table 1 Medicaid patients represent 33%. Why is this the same proportion for all diseases? And what explains the difference relative to the national average of 3%?).

5. Discussion: Based on the previous comment, I think it is crucial to provide a detailed discussion on the channel that affects the access to care. The combination of the disease mix (which you already have) with the overall number of admissions by SES (which I suggested in the previous comment) will allow you to provide additional insights. For instance, are we observing a decrease in admissions with only the highest risks being selected (and a crowding out of the lowest risks)? Is the decrease in admissions greater in lower SES? And, in relative terms, is the crowding out of lower risks effect larger in lower SES?

6. Conclusion: this section should summarize your main results. I would expect to see what was the impact of the pandemic on the access to care for both groups of SES. However, you present your main result as “Patients with CRS and oral ulcers presented with higher disease severity at the time of first hospital visit during the COVID-19 pandemic pared to the pre-pandemic era.”. This is interesting, but is not an answer to your research question. Mprevver, you mention that “the COVID-19 outbreak seems to affect the previously exiting healthcare access discrepancy between different socioeconomic status”. This is more relevant and aligned with your research question. You should be more clear on the message and explicitly mention the direction of the effect. “seems to affect” in which direction? Does this reinforce the pre-existing inequality? Why?

Additionally, some minor could be incorporated to further improve the paper:

1. Data on Socioeconomic Status: please provide some brief discussion in the paper on the quality of the proxy that you have used for SES. Only 3% of Koreans are enrolled in Medicaid programs. This seems a very low number to represent “low socio-economic status” – and should only represent the very very poor. Isn’t there an alternative to distinguish other groups? Moreover, how reliable is the inclusion of patients to each program? Is there potential for fraud or misallocation? Please provide a critical perspective on the potential limitations (and benefits) of using this variable as a proxy.

2. Institutional background: following the last comment, it would be useful for the reader to see some brief description of the institutional background of the Korean Health System. This description should not only include an explanation of the insurance mechanisms, but also a description of the patient pathways. In particular, are there GPs acting as gatekeepers? Or can a booking of an otorhinolaryngology appointment be done directly by the patient? This is important to understand where are the access constraints that may have been influenced by the pandemic.

3. In the text, you should refer to table 2.

I hope you find these comments useful and that they can contribute to an improved version of your paper.

All the best

Author Response

Response to Reviewer 1 Comments

  1. Geography and External Validity: there is no indication in the abstract and in the introduction on the country / setting in which this analysis is being performed. This should be clear since the beginning of the paper – eventually, even in the title. Moreover, it should be discussed how are these results generalizable (or not) to other countries and other types of health systems: a discussion on the external validity of these results in missing in the paper.

 Thank you for your comment. This study included patients who visited a metropolitan public hospital in Seoul, Korea between February 2018 to February 2021 and were diagnosed with chronic rhinosi-nusitis (CRS), sensorineural hearing loss (SNHL), and oral ulcers. The patients’ medical insurance status, age, gender, audiome-try results, endoscopic findings and imaging findings were reviewed. We have mentioned the above contents in the introduction section as below:

“In this study, Otorhinolaryngology patients’ disease severity status at the first time of their hospital visit and there medical health insurance status were investigated during the pre -and post-COVID-19 pandemic era in a single medical center located in Seoul, Korea.”

We found that during the COVID-19 pandemic, while the total number of outpatients was reduced, the severity of these Otorhinolaryngology diseases seemed to increase compared to the pre-pandemic severity in every SES group. We assume that during the pandemic, factors such as fear of disease transmission and quarantine may have prevented people from seeking medical care early stages of their disease. We have discussed the external validity of these results and how we can generalize the results in section of discussion as below:

“Therefore, generally lowering the barrier to health care service across all socioeconomic groups should also be considered in public health polices during the era of pandemics.”

“It is essential to maintain efforts to provide fundamental medical services during the pandemic, especially with at-risk populations, in order to improve overall public health.”

  1. Contribution: you claim that the concerns regarding patient access to healthcare during the pandemic have been already reported in the literature, with many different perspectives. Given this, and based on the paper, it is not clear what is the contribution of this paper to the current literature. I suggest you to clarify this in advance in the introduction (after referring to the existing literature) and in the discussion section. Clarifying your contribution, will help you “selling” your paper! In particular, it would be great to discuss why is it relevant to look at otorhinolaryngology patients. Do we expect results for this particular group of patients to be different? Why? What makes them special?

Thank you for your kind suggestion. We agree with the importance of clarifying the contribution and discussing why is it relevant to look at otorhinolaryngology patients, especially during the pandemic. As you suggested we have discussed the clinical importance of otorhinolaryngology disease in the section of discussion as below:

“Health disparities in the field of otorhinolaryngology are of particular interest, given that these problems can have a direct negative impact on quality of life due to functional impairment of the head and neck region. Hence, otorhinolaryngology diseases tend to be relatively more sensitive to factors that limit access to healthcare.”

  1. Research Question: I feel that the way you present your results is disconnected from your original research question. You claim that “The aim of this study was to investigate the impact of COVID-19 on access to healthcare for otorhinolaryngology patients from different socioeconomic status (SES) groups”. However, you present your results in terms of changes in the severity mix. You do not explicitly compare the access to care for the two groups – only the change in severity for the two groups. Thus, I advise you to rewrite your results, discussion and conclusion and align them more closely with your original (and relevant) research question. My next comments provide some suggestions to do this.

Thank you for your comment. We agree that comparing the disease severity status at the first hospital visit may not represent the whole health care accessibility. However, delayed access to the healthcare system has been assumed to negatively affect health outcomes due to delays in diagnosis and treatment. We tried to explain this issue in the section of methods as below:

In this study, healthcare accessibility was estimated by measuring the disease sever-ity diagnosed by an otorhinolaryngology physician at the patient’s first outpatient clinic visit, following the assumption that lower SES can disturb and reduce healthcare-seeking behavior, leading to less access to healthcare, and consequently causing a delayed appearance to the clinic

  1. Results: you are mostly looking to severity mix changes and not to access to care. It would be extremely relevant to complement this with overall number of admissions. You provide the overall number of admissions for the total sample, but you do not distinguish this between groups. I suggest to i) provide additional information on the number of visits by groups of SES; ii) consider replacing figure 1 by a table and test whether the differences between groups are statistically significant. Include the information of the number of visits by groups in this table; iii) provide an explanation for the high volume of Medicaid patients presented in table 1 (earlier you claim that, on average, Medicaid patients represent 3% of the population. However, in table 1 Medicaid patients represent 33%. Why is this the same proportion for all diseases? And what explains the difference relative to the national average of 3%?).

Thank you for your question. As you suggested, we’ve added the table regarding total number of patients to assess the number of 1st visits per month before and after the pandemic. We found that lower SES group patients’ number of 1st visits per month was decreased after the pandemic, but higher SES group patients’ hospital visits decreased as well. We added the additional results as below:

3.2. Number of 1st vist patients during Peri-COVID-19 period

In every disease group, the number of first-time patient visits decreased during the COVID-19 pandemic as compared to the number of first-time patient visits before the pandemic. In the CRS group, 31.9 patients first visited the hospital before the pandemic, and the number decreased to 17.1 patients per month after the pandemic. In the SNHL group, 51.9 patients per month first visited the hospital before the COVID-19 pandemic, then the number declined to 42.1 patients per month during the peri-COVID 19 period. The number of patients who presented with oral ulcers declined from 5.46 to 4.05 patients per month after the outbreak. (Table 1)

 In our study, for each disease, Medicaid group was compared with age - and sex-matched controls who were randomly selected from the National Health Insurance patient group. Therefore the proportion was 2:1 since they were selected as a double-numbered group for the comparison. We’ve done the selection since the Medicaid group was relatively small (3%) as you’ve noted.

  1. Discussion: Based on the previous comment, I think it is crucial to provide a detailed discussion on the channel that affects the access to care. The combination of the disease mix (which you already have) with the overall number of admissions by SES (which I suggested in the previous comment) will allow you to provide additional insights. For instance, are we observing a decrease in admissions with only the highest risks being selected (and a crowding out of the lowest risks)? Is the decrease in admissions greater in lower SES? And, in relative terms, is the crowding out of lower risks effect larger in lower SES?

Thank you for your comment. The overall disease severity at the time of first visit was increased after the pandemic, and this was probably due to the assumption that people who were not gravely affected disease chose not to visit the clinic. We’ve added about the assumption in the context.

Notably for CRS patients, the impact of SES over disease severity decreased after COVID-19 pandemic, which might be the result of general avoidance of healthcare visit due to the fear of pandemic, especially when people were expected to contact with other patients with upper respiratory symptoms. Also, before the pandemic, peo-ple who were insured with Medicaid might have sought healthcare less frequently than those with National Health Insurance did, and presented with higher disease se-verity on their first hospital visit. However, during the pandemic, even people who were insured with National Health Insurance, used healthcare systems much less than before, resulting in a higher disease severity on their first visit

Lazzerini M.; Barbi E.; Apicella A.; et al. Delayed access or provision of care in Italy resulting from fear of COVID-19. The Lancet Child & Adolescent Health 2020, 4 e10-e11.

  1. Conclusion: this section should summarize your main results. I would expect to see what was the impact of the pandemic on the access to care for both groups of SES. However, you present your main result as “Patients with CRS and oral ulcers presented with higher disease severity at the time of first hospital visit during the COVID-19 pandemic pared to the pre-pandemic era.”. This is interesting, but is not an answer to your research question. Moreover, you mention that “the COVID-19 outbreak seems to affect the previously exiting healthcare access discrepancy between different socioeconomic status”. This is more relevant and aligned with your research question. You should be more clear on the message and explicitly mention the direction of the effect. “seems to affect” in which direction? Does this reinforce the pre-existing inequality? Why?

Thank you for your kind comment. The COVID-19 outbreak seems to affect the previously exiting healthcare access discrepancy between different socioeconomic status groups with various factors. We’ve revised the conclusion as below:

“Within CRS and SNHL group, Medicaid group tends to have higher disease severity compared to National Health Insurance group, and interestingly the OR decreased after COVID-19 pandemic in CRS group. However in SNHL group, the discrepancy be-tween two groups severity increased after COVID-19 pandemic although not statistically significant.”

Additionally, some minor could be incorporated to further improve the paper:

  1. Data on Socioeconomic Status: please provide some brief discussion in the paper on the quality of the proxy that you have used for SES. Only 3% of Koreans are enrolled in Medicaid programs. This seems a very low number to represent “low socio-economic status” – and should only represent the very very poor. Isn’t there an alternative to distinguish other groups? Moreover, how reliable is the inclusion of patients to each program? Is there potential for fraud or misallocation? Please provide a critical perspective on the potential limitations (and benefits) of using this variable as a proxy.

Thank you for your question. The SES level information was obtained through the medical records which provides the information of the payment method. In Korea, the general population is divided into 4 categories according to their socioeconomic status level. Patients pay their medical expense according to these categories which are embedded in the electric medical recording system. Therefore the possibility of forgery or fraud might expected to be low, but potential limitations of using this variable might include that this variable might not reflect multiple elements of socioeconomic status of a person.

  1. Institutional background: following the last comment, it would be useful for the reader to see some brief description of the institutional background of the Korean Health System. This description should not only include an explanation of the insurance mechanisms, but also a description of the patient pathways. In particular, are there GPs acting as gatekeepers? Or can a booking of an otorhinolaryngology appointment be done directly by the patient? This is important to understand where are the access constraints that may have been influenced by the pandemic.

Thank you for the detailed comment. As you’ve noted, to visit a secondary or tertiary medical center in Korea, an individual should be referred from a local practitioner, which could be another blockage to seek advanced healthcare in the time of pandemic. We added this description in the discussion part as follows:

“Furthermore, to visit a secondary or tertiary medical center in Korea, an individual should be referred from a local practitioner, which could be another blockage to seek advanced healthcare in the time of pandemic”

  1. In the text, you should refer to table 2.

The referral was added in the Results section. Thank you.

Reviewer 2 Report

The article poses an interesting questions but please consider the following suggestions to improve it. Moreover, please consider using a different design to analyze your data, also relying on the opinion of an expert modeller. 

1) abstract: please define all abbreviations (CRS, ENT)

2) abstract: please specify where the study was conducted and the type of data collection performed

3) abstract: please provide also some quantitative findings

4) methods: specify that cancer was not included among the diseases considered.

5) methods: you might consider to adopt a regression discontinuity design, using the COVID-19 outbreak as the cut-off in the time variable (x-axis) and comparing the pre-pandemic data with the pandemic data (y-axis). 

6) Table 1: please report % in brackets besides absolute numbers and add numbers for grading

7) Table 2: please specify the type of regression performed

8) Discussion: please include any policy implications for your country

Author Response

=

Response to Reviewer 2 Comments

Point 1: The article poses an interesting questions but please consider the following suggestions to improve it. Moreover, please consider using a different design to analyze your data, also relying on the opinion of an expert modeller.

1)             abstract: please define all abbreviations (CRS, ENT)

Thank you for your kind comment. The abbreviations were defined.

2)             abstract: please specify where the study was conducted and the type of data collection performed

Thank you for your suggestion. We have specified the detail information of where this study was conducted and the type of data collection performed in section of methods as below:

“Otorhinolaryngology patients’ disease severity status, diagnosed at the first hospital visit, was investigated during the pre -and post-COVID-19 pandemic era in a single medical center located in Seoul, Korea.”

3) abstract: please provide also some quantitative findings

Thank you for your suggestion. We have mentioned the total number of patients and the odds ratio in the abstract.

4) methods: specify that cancer was not included among the diseases considered.

Thank you for your comment. The exclusion was specified.

5) methods: you might consider to adopt a regression discontinuity design, using the COVID-19 outbreak as the cut-off in the time variable (x-axis) and comparing the pre-pandemic data with the pandemic data (y-axis).

Thank you for your kind suggestion. We tried to assess the impact of COVID-19 within homogenous SES group patients during the peri-pandemic period as suggested. Moreover, we tried to evaluate the impact of SES within a certain period (pre-or post-pandemic) of time, to see if the impact of SES changes. Therefore, we prefer to use the generalized estimating equation method as previously reported.

6) Table 1: please report % in brackets besides absolute numbers and add numbers for grading

Thank you for your comment. We’ve repositioned tables and numbers were added.

7) Table 2: please specify the type of regression performed

Generalized estimating equation (GEE) method for matched ordinal multinomial was performed, and added in the Table.

8) Discussion: please include any policy implications for your country

Thank you for your kind suggestion. We think that a systematic method to assess and respond to exacerbated healthcare inequity, as well as informed distribution of healthcare assets during the time of pandemic are necessary to our country’s healthcare system. We have discussed the external validity of these results and how we can generalize and imply the results in section of discussion as below:

“Therefore, generally lowering the barrier to health care service across all socioeconomic groups should also be considered in public health polices during the era of pandemics.”

“It is essential to maintain efforts to provide fundamental medical services during the pandemic, especially with at-risk populations, in order to improve overall public health.”

We all sincerely thank the reviewers for the comments and their precious time again.

Reviewer 3 Report

[1] Please add references and give a reader friendly explanation for your assumption that lower SES can disturb and reduce healthcare-seeking behavior. In this manuscript the key mechanism that lower SES people were less likely to select physician visits is unclear. 

[2] Please add a detailed explanation of the empirical strategy. I think most readers cannot understand that the GEE method for matched ordinal multinomial is the best strategy when analyzing the impact of COVID-19 on access to healthcare.

Author Response

Response to Reviewer 3 Comments

[1] Please add references and give a reader friendly explanation for your assumption that lower SES can disturb and reduce healthcare-seeking behavior. In this manuscript the key mechanism that lower SES people were less likely to select physician visits is unclear.

 Thank you for your comment. The ease of access to healthcare has become an issue of social equity. Inequalities in access to healthcare across different population groups are known to be the main reason for existing disparities. Therefore, in this paper we tried to assess the correlation between SES and access to care which was represented as disease severity at the point of first hospital visit, since we have hypothesized that lower SES might limit access to healthcare and, in turn, contribute to large social inequalities. We agree that comparing the disease severity status at the first hospital visit may not represent the whole health care accessibility. However, delayed access to the healthcare system has been assumed to negatively affect health outcomes due to delays in diagnosis and treatment. We have added this issue with references in the section of discussion as below:

In this study, healthcare accessibility was estimated by measuring the disease severity diagnosed by an otorhinolaryngology physician at the patient’s first outpatient clinic visit, following the assumption that lower SES can disturb and reduce healthcare-seeking behavior, leading to less access to healthcare, and consequently causing a delayed appearance to the clinic

Andersen R.; Aday LA. Access to medical care in the U.S.: realized and potential. Medical Care 1978,16, 533-546.

Jabbour J.; Robey T.; Cunningham MJ. Healthcare disparities in pediatric otolaryngology: A systematic review. The Laryngoscope 2018, 128, 1699-1713.

Talwar A.; Sahni S.; Talwar A.; Kohn N.; Klinger J. Socioeconomic Status Affects Pulmonary Hypertension Disease Severity at Time of First Evaluation. Pulmonary Circulation 2016, 6, 191-195.

 [2] Please add a detailed explanation of the empirical strategy. I think most readers cannot understand that the GEE method for matched ordinal multinomial is the best strategy when analyzing the impact of COVID-19 on access to healthcare.

Thank you for your question. We have tried to assess the impact of COVID-19 on existing healthcare inequity in South Korea, which was suggested in our previous study. We tried to assess the impact of COVID-19 within homogenous SES group patients during the peri-pandemic period. Also, we tried to estimate the impact of SES within a certain period (pre-or post-pandemic) of time, to see if the impact of SES changes. Therefore, we have used the generalized estimating equation method.

Hong SN.; Kim JK.; Kim DW. The Impact of Socioeconomic Status on Hospital Accessibility in Otorhinolaryngological Disease in Korea. Asia Pacific Journal of Public Health 2021, 33, 287-292.

We all sincerely thank the reviewers for the comments and their precious time again

Round 2

Reviewer 1 Report

Dear authors,

Thank you for your reply. At this point, I have no further comments regarding your manuscript.

Best regards

Author Response

Thank you for your detailed review. We all sincerely thank the reviewers for the comments and their precious time again.

Reviewer 2 Report

Thanks for addressing my previous comments.

Author Response

Thank you for your comments. We all sincerely thank the reviewers for the comments and their precious time again

Reviewer 3 Report

Please add a detailed explanation of the empirical strategy. I think most readers cannot understand that the GEE method for matched ordinal multinomial is the best strategy when analyzing the impact of COVID-19 on access to healthcare.

Author Response

Thank you for your question. We have used the generalized estimated equation since the severities of diseases at the time of first hospital visit were ordinal categorized data, and we have to assess longitudinal changes of the ordinal data. We have added the suggested content to the manuscript as follows:

“The severity of diseases at the time of first hospital visit were examined, and the longitudinal change in disease severity was assessed for each group with ordinal regression analyses.”

Thank you for the comments and your precious time again
